# Regional and Urban-Scale Environmental Influences of Oceanic DMS Emissions over Coastal China Seas

**Shanshan Li [1], Yan Zhang [1,2,3,], Junri Zhao [1] , Golam Sarwar [4], Shengqian Zhou [1] , Ying Chen [1], Guipeng Yang [5] and Alfonso Saiz-Lopez [6]**

1   Shanghai Key Laboratory of Atmospheric Particle Pollution and Prevention (LAP3),
    Department of Environmental Science and Engineering, Fudan University, Shanghai 200438, China;
    shanshan_li1130@163.com (S.L.); 19110740023@fudan.edu.cn (J.Z.); 17110740002@fudan.edu.cn (S.Z.);
    yingchen@fudan.edu.cn (Y.C.)
2   Big Data Institute for Carbon Emission and Environmental Pollution, Fudan University,
    Shanghai 200433, China
3   Institute of Atmospheric Sciences, Fudan University, Shanghai 200438, China
4   Center for Environmental Measurement and Modeling, Office of Research and Development,
    U.S. Environmental Protection Agency, Research Triangle Park, NC 27711, USA; Sarwar.Golam@epa.gov
5   Key Laboratory of Marine Chemistry Theory and Technology, Ministry of Education, College of Chemistry
    and Chemical Engineering, Ocean University of China, Qingdao 266100, China; gpyang@ouc.edu.cn
6   Department of Atmospheric Chemistry and Climate, Institute of Physical Chemistry Rocasolano, CSIC,
    28006 Madrid, Spain; a.saiz@csic.es
*   Correspondence: yan_zhang@fudan.edu.cn; Tel.: +86-136-3637-2289

**Abstract:** Marine biogenic dimethyl sulfide (DMS) is an important natural source of sulfur in the atmosphere, which may play an important role in air quality. In this study, the WRF-CMAQ model is employed to assess the impact of DMS on the atmospheric environment at the regional scale of eastern coastal China and urban scale of Shanghai in 2017. A national scale database of DMS concentration in seawater is established based on the historical DMS measurements in the Yellow Sea, the Bohai Sea and the East China Sea in different seasons during 2009~2017. Results indicate that the sea-to-air emission flux of DMS varies greatly in different seasons, with the highest in summer, followed by spring and autumn, and the lowest in winter. The annual DMS emissions from the Yellow Sea, the Bohai Sea and the East China Sea are 0.008, 0.059, and 0.15 Tg S a$^{-1}$, respectively. At the regional scale, DMS emissions increase atmospheric sulfur dioxide ($SO_2$) and sulfate ($SO_4^{2-}$) concentrations over the East China seas by a maximum of 8% in summer and a minimum of 2% in winter, respectively. At the urban scale, the addition of DMS emissions increase the $SO_2$ and $SO_4^{2-}$ levels by 2% and 5%, respectively, and reduce ozone ($O_3$) in the air of Shanghai by 1.5%~2.5%. DMS emissions increase fine-mode ammonium particle concentration distribution by 4% and 5%, and fine-mode nss-$SO_4^{2-}$ concentration distributions by 4% and 9% in the urban and marine air, respectively. Our results indicate that although anthropogenic sources are still the dominant contributor of atmospheric sulfur burden in China, biogenic DMS emissions source cannot be ignored.

**Keywords:** DMS emissions; WRF-CMAQ; eastern China Seas; Shanghai; atmospheric environment

## 1. Introduction

The global emissions and environmental effects of biogenic sulfur have become increasingly important with the decline of anthropogenic sulfur dioxide ($SO_2$) emissions. Dimethylsulfide (DMS: $CH_3SCH_3$) is the most abundant volatile biogenic sulfide emanating from the ocean [1,2], which is

the crucial precursor for atmospheric sulfate aerosol and cloud condensation nuclei in the marine boundary layer and thus has profound climate implications [3].

The potential importance of DMS in climate regulation provides extensive motivation for an accurate representation of its surface ocean concentrations. The concentration of DMS in seawater may vary by orders of magnitude in time and space, ranging from $10^{-2}$ to $10^2$ nmol/L. The seasonal cycle of seawater DMS concentrations shows a maximum in summer and minimum in winter at mid-latitudes and high latitudes [4–7]. According to Aranami et al. [8], the average concentration of DMS in summer can even be about 5 times that of autumn and winter in the North Pacific. However, the concentration of DMS in the surface seawater in the equatorial region is substantially invariant with season [9,10]. As for the distribution in the sea area, the DMS concentration appears to be higher in high-productivity areas such as the continental shelf, offshore, and estuaries than in remote oceanic areas. For example, Andreae [11] estimated that the average concentration of DMS in the global ocean was 0.3 nmol $L^{-1}$, while Turner et al. [12] found that the average concentration in seawater can reach 7.0 nmol $L^{-1}$ on the East Anglian coast. On the basis of multiple efforts in sea surface DMS measurements, a global DMS database was compiled as a key tool for DMS flux estimates [7]. Lana et al. [13] updated the results with a threefold increase of data (from 15,000 to over 47,000) in the Global Surface Seawater DMS Database (GSSDD). This database, which is currently maintained at the NOAA-PMEL (National Oceanic and Atmospheric Administration-Pacific Marine Environmental Laboratory), continues to be updated and freely available to the scientific community. However, the current observational data are sparsely distributed in both space and time, especially in China's waters. The DMS data are still based on Yang et al.'s study in the South China Sea in 1993 [14] and in the East China Sea in 1994 [15], which brings great uncertainty to DMS simulation work. Therefore, it is necessary for the establishment of a localized DMS concentration database in China.

Accurate estimation of DMS sea-to-air emission flux is critical to understanding the global bio-sulfur cycle and its impact on the radiation budget. The sea-to-air flux of DMS is dependent on the DMS concentration in seawaters and the sea-to-air transfer velocity which varies with sea surface temperature (SST) and wind speed [16–18]. The preliminary estimated global DMS emission flux was about $40 \pm 20$ Tg S $a^{-1}$ by Andreae et al. [19]. Bates et al. [2,20] pointed out that the above estimates may be too high using DMS measurements in summer when productivity was high. Dividing the year into two periods of summer and winter and taking into account the spatial differences in DMS concentrations, they reported estimates of 16 Tg S $a^{-1}$. In a subsequent study, Putaud and Nguyen [21] reported annual global emissions estimates of 17–21 Tg S $a^{-1}$, Lana et al. [13] reported estimates of 17.6–34.4 Tg S $a^{-1}$, and Tesdal et al. [22] reported estimates of 18–24 Tg S $a^{-1}$. The Chinese research on DMS sea-to-air emission flux mainly focuses on on-site observations in certain areas of the Yellow Sea, the Bohai Sea, the East China Sea and the South China Sea [23–26]. Yang et al. [27] estimated the emission flux to be 5.32–11.92 $\mu$mol $m^{-2}$ $d^{-1}$, and the annual DMS emission was $7.41 \times 10^{-2}$–$16.60 \times 10^{-2}$ Tg S $a^{-1}$ in the East China Sea and the Yellow Sea based on the DMS concentration and wind speed data obtained from the cruise survey during 2006–2007. The Yellow Sea, the Bohai Sea, and the East China Sea are among the largest continental shelf seas in the world, which can contribute to higher DMS concentrations owing to higher productivity levels. Therefore, although China's continental shelf area accounts for only a small portion of the world's oceans, the contribution to global marine DMS emissions may not be negligible.

Numerical modeling plays an important role in studying the impact of DMS on regional and urban atmospheric environments. Large-scale atmospheric chemistry models usually contain very simplified DMS chemistry, including the oxidation of DMS by hydroxyl radical (OH), nitrate ($NO_3$) to directly produce $SO_2$ and methanesulfonic acid (MSA), while ignoring the contribution of halogen to DMS oxidation [28–32]. Sarwar et al. [33] incorporated halogen chemistry, the calculation of DMS sea-to-air emission fluxes and simplified DMS chemistry into the hemispheric CMAQ (Community Multiscale Air Quality model. After the addition of DMS chemistry, the $SO_2$ concentration over many sea areas increased by more than 40 parts per trillion by volume (pptv), and the annual average

concentration of $SO_4^{2-}$ increased by more than 0.8 µg/m$^3$. Due to the high DMS emissions and high oxidant concentration, DMS showed the greatest impact in summer with the average concentration of $SO_4^{2-}$ increasing by 70%. At the city scale, Muñiz-Unamunzaga et al. [34] performed a high-resolution (4 km×4 km) simulation of Los Angeles using the CMAQv5.1 model. The marine emissions of halogens and DMS led to major changes in the concentration levels of atmospheric oxidants (such as OH, HO$_2$ (hydroperoxy radical), and NO$_3$) and the composition and mass of fine particles. Although the concentration of ozone (O$_3$), NO$_3$, and HO$_x$ decreased, the average concentration of secondary organic aerosol (SOA) increased by 10%. Perraud et al. [35] applied the UCI-CIT (University of California Irvine–California Institute of Technology) airshed model to study atmospheric concentrations in the South Coast Air Basin. They included emissions of organosulfur compounds from ocean, agricultural activities, and urban sources and reported that particles can continue to be formed in the coastal urban atmosphere from the oxidation of organosulfur compounds even when anthropogenic SO$_2$ emissions are reduced to zero. In China, DMS-related studies generally focused on field observation. There have been few reports on the impact of DMS in China seawater on the regional and urban atmospheric environment using numerical modeling.

The main objective of this study is to develop a localized database of DMS concentrations in seawater for estimating DMS fluxes in China and examine the impacts of seasonal DMS emissions on regional and urban atmospheric environment. This paper aims to accumulate experience for the study of the marine biogenic sulfur cycle in China and lay the foundation for further exploration of the climatic effects of DMS.

## 2. Methodology

### 2.1. Model Description

The US EPA's CMAQ modeling system version 5.2 was used in this study. The CMAQ v5.2 model was configured to utilize the CB05 gas phase chemistry mechanism together with halogen chemistry (cb05eh51) [36,37]. DMS chemistry was added to the model based on the study of Sarwar et al. [33] which includes the oxidation of DMS by OH, NO$_3$, bromine monoxide (BrO), chlorine radical (Cl), chlorine monoxide (ClO) and iodine monoxide (IO). The detailed reactions are provided in Table S1. The model was configured to use the sixth-generation aerosol module (AERO6), which includes a comprehensive treatment of inorganic aerosols, organic aerosols including secondary organic aerosol (SOA), dust, trace metals, sea salt, and unspeciated material [38,39].

High-resolution simulations were performed for three nested model domains on a Lambert conic conformal projection (Figure 1). The outermost domain was developed with 115 × 163 horizontal grids and a resolution of 36 km, covering all of East Asia. The intermediate domain covers the eastern part of China using a 12 km × 12 km horizontal grid-resolution, and the innermost domain covers all of Shanghai using a finer 4 km × 4 km horizontal grid-resolution. Four months in the year 2017, January, April, July, and October, which represent winter, spring, summer, and autumn, respectively, were selected to compare the seasonal effects. The static initial condition was used with a 15-day model spin-up period to minimize the effect of initial condition on model predictions. Two simulations were performed for each model domain: one without DMS emissions and the other with DMS emissions. The difference between the two simulations is attributed to DMS emissions. Boundary conditions without and with DMS emissions for the 36-km simulation were generated from the corresponding hemispheric CMAQ model results [33]. The meteorological field required for the CMAQ model was generated by the Weather Research and Forecasting (WRFv3.6) model [40] and processed by the MCIPv4.3 (Meteorology-Chemistry Interface Processor) [41]. The WRF model was initialized from the National Centers for Environmental Prediction/the National Center for Atmospheric Research reanalysis data with a resolution of 1° × 1° spatially and 6 h intervals temporally. Vertically, 27 sigma layers were used for the WRF-CMAQ simulations.

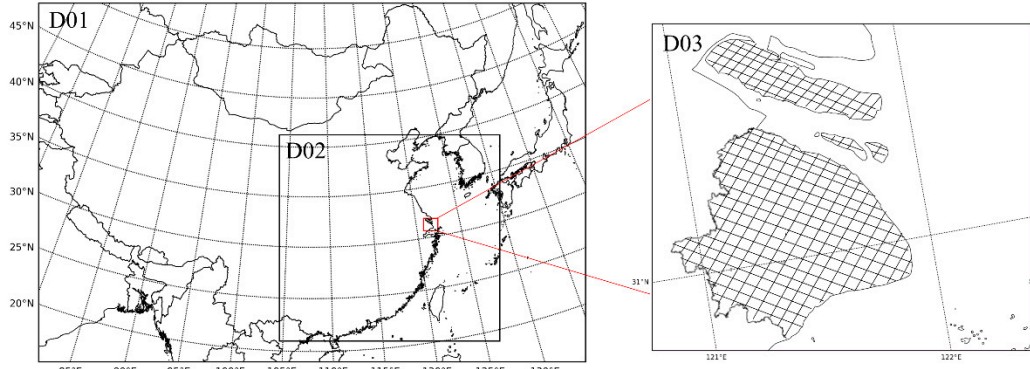

**Figure 1.** The modeling domains of WRF-CMAQ (Weather Research and Forecasting-Community Multiscale Air Quality). The outermost domain covers all of East Asia, the intermediate domain covers the eastern part of China, and the innermost domain covers the whole of Shanghai. (The urban areas of Shanghai are marked with grid).

### 2.2. Emissions Inventories

In this study, anthropogenic emissions for the outermost domain were obtained from the Multi-Resolution Emission Inventory for China (MEIC v1.3) developed by Tsinghua University and MIX inventory developed by the Model Inter-Comparison Study for Asia (MICS-Asia III) and Hemispheric Transport of Air Pollution (HTAP) for other areas. The most recent emission inventory data is 2016 for MEIC and 2010 for MIX [42], which contains the gridded data of ten major air pollutants and greenhouse gases ($PM_{2.5}$, $PM_{coarse}$, $SO_2$, $NH_3$, $NO_x$, BC, OC, CO, $CO_2$, and VOC). The terrestrial anthropogenic emissions inventory in the Yangtze River Delta in 2017 was provided by the Shanghai Academy of Environmental Sciences (SAES) at a 4 km × 4 km resolution and included nine pollutants ($PM_{2.5}$, $PM_{10}$, $SO_2$, $NH_3$, $NO_x$, VOC, CO, $CO_2$, and $CH_4$). Since DMS is a marine biogenic sulfur gas, in addition to the above land-based emission inventories, anthropogenic emissions, especially anthropogenic sulfur emissions over the ocean areas, are also of great importance. The shipping emissions were then constructed using a model developed by Fan et al. [43] based on the automatic identification system (AIS) data of 2017 supplied by the Shanghai International Shipping Research Center. The detailed principle and methods have been previously described by Feng et al. [44]. The combined inventory showed that the $SO_2$ emissions generated by human activities in January, April, July, and October were 2.3 Tg, 1.6 Tg, 1.8 Tg and 1.8 Tg, respectively, among which shipping emissions accounted for 46 Gg, 56 Gg, 63 Gg, and 62 Gg, respectively. Marine emissions of halogen species include two inorganic iodine species (HOI and I2) and nine halocarbons ($CH_3I$, $CH_2ICl$, $CH_2IBr$, $CH_2I_2$, $CHBr_3$, $CH_2Br_2$, $CH_2BrCl$, $CHBrCl_2$, $CHBr_2Cl$). The details of halogen emissions have been described elsewhere [36,37].

### 2.3. Database of Observed DMS Concentrations in Seawater

For model studies, the concentration data of DMS in seawater from the Global Surface Seawater DMS Database are generally used. The spatial and temporal distribution of DMS concentration in seawater from this global database are shown in Figure S1. The concentration of DMS in seawater presents a decreasing trend from nearshore to remote oceans. The DMS concentration is higher in spring and summer, up to 5.95 nmol/L, followed by autumn, and is the lowest in winter. However, the DMS historical observation data in Chinese waters in the global database is relatively old with the coarse resolution, which may be incompatible with the current conditions and could lead to unrealistic DMS emission flux estimates. We obtained a long term DMS observation data set from 2009-2017 from a series of cruise survey experiments led by the Ocean University of China [24,25,45–53]. Due to the lack of observation data in the South China Sea, the observation data covers mainly the Yellow Sea, the Bohai Sea, and the East China Sea. The summary of historical cruise surveys is shown in

Table S1. The number of cruise surveys in summer is the largest. The multi-year observation data are interpolated by ordinary Kriging [54] and the average of DMS concentration in different cruise surveys in each season is used to map DMS concentrations in Chinese seas ranging from 23.7° N to 40.3° N, 118.2° E to 128.0° E. The data from the global DMS database is still used in the areas not covered by the cruise surveys. The spatial distribution of DMS concentration in seawater for four seasons is shown in Figure 2. The average concentration of DMS in seawater is the highest in summer, which can reach 20.8 nmol/L in the Bohai Sea and East China Sea; followed by spring with the maximum in the East China Sea at 10.7 nmol/L and autumn with the maximum at 6.4 nmol/L, and the lowest in winter when the maximum value of DMS concentration is only 2.6 nmol/L. The average concentration of DMS in the entire simulated sea area in spring, summer, autumn, and winter based on the localized DMS database is 1.89, 1.23, 1.25, and 1.50 times, respectively, higher than the global DMS database. This reveals the significant underestimation of DMS concentration in China's seawaters from the global database and emphasizes the need for a localized DMS database.

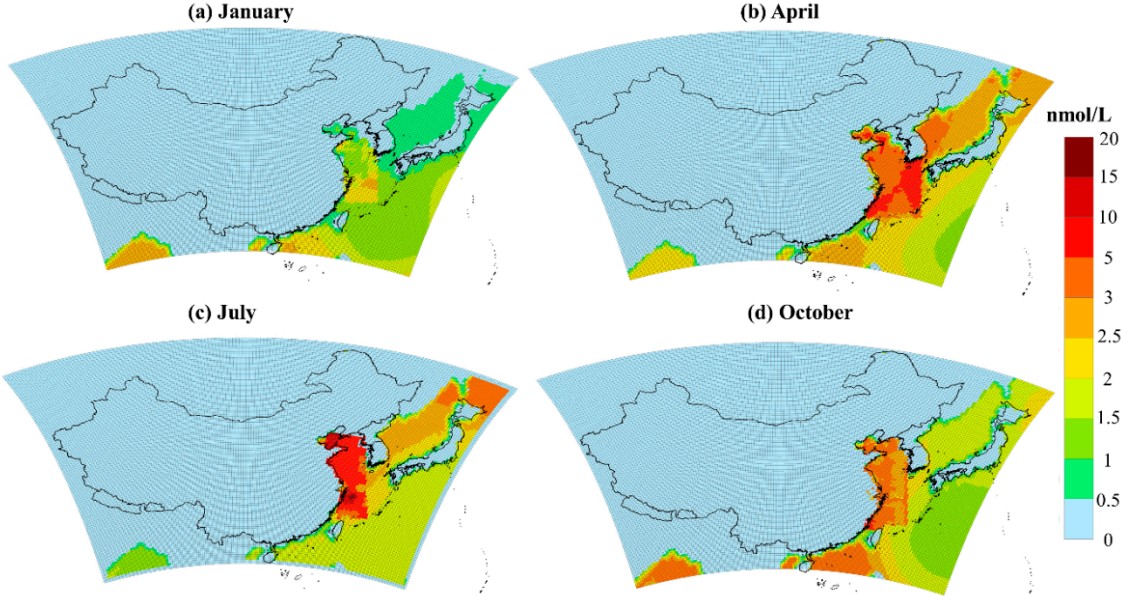

**Figure 2.** A national scale database of DMS (dimethyl sulfide) concentration—a multi-year observational record in the East China seas including the Yellow Sea, the Bohai Sea, and the East China Sea, and the data from the global DMS database is still used in the areas not covered by the observations. (**a**), (**b**), (**c**), and (**d**) represent the concentration distribution of DMS in seawater in January, April, July and October, respectively.

*2.4. Parameterizations of Air-Sea Gas Exchange*

Ocean-atmosphere DMS fluxes are computed as the product of the concentration gradient between air and water ($\Delta c$) and the total gas transport resistance at the air/ocean interface ($k_T$) [13], as follows:

$$F_{DMS} = k_T \times \Delta c = k_T \times (C_w - C_g \times H) \approx k_T \times C_w \qquad (1)$$

where $C_w$ is the concentration of DMS in seawater, $C_g$ is the concentration of DMS in air, H is Henry's law coefficient. The DMS concentration in seawater is about three orders of magnitude higher than that in air, so the $C_g \times H$ term can generally be ignored. The total gas transmission rate $k_T$ can be further obtained by the water side transmission rate ($k_w$) and atmospheric gradient fraction ($\gamma_a$) [55]:

$$k_T = k_w \times (1 - \gamma_a) \qquad (2)$$

$$\gamma_a = 1/(1 + k_a/Hk_w) \qquad (3)$$

The air side transmission rate $k_a$ can be calculated according to Kondo [56]:

$$k_a = 659 \times U_{10}/\sqrt{(M_{DMS}/M_{H2O})} \tag{4}$$

where $U_{10}$ is 10-meter wind speed, $M_{DMS}$ and $M_{H2O}$ are the molecular weights of DMS and water, respectively. It is the parameterization of $k_w$ that the largest uncertainty in DMS emission flux comes from. In this study, the Liss and Merlivat, 1986 (LM86) [16] parameterization is used in the model to calculate the water-side ($k_w$) transfer velocity:

$$k_w = \begin{cases} 0.17 \times U_{10} / (Sc_{DMS}/600)^{2/3} & U_{10} \leq 3.6 \ m/s \\ (2.85 \times U_{10} - 9.65) / (Sc_{DMS}/600)^{1/2} & 3.6 < U_{10} \leq 13 \ m/s \\ (5.9 \times U_{10} - 49.3)/(Sc_{DMS}/600)^{1/2} & U_{10} > 13 \ m/s \end{cases} \tag{5}$$

where $Sc_{DMS}$ is the Schmidt number of DMS related to the sea surface temperature SST (T/°C) [57]:

$$Sc_{DMS} = 2674.0 - 147.12 \times T + 3.726 \times T^2 - 0.038 \times T^3 \tag{6}$$

## 3. Results and Discussion

### 3.1. Evaluation of Model Results

Model performance for the regional domain is evaluated by comparing model predictions with observations of atmospheric $SO_2$ and $PM_{2.5}$ (fine particles) concentrations from nine eastern coastal cities (from south to north: Zhuhai, Xiamen, Ningbo, Zhoushan, Shanghai, Nantong, Qingdao, Tianjin, Qinhuangdao). The statistical metrics of normalized mean bias (NMB), root mean-square error (RMSE) and correlation coefficient (r) are calculated to quantify the degree of deviation between the observed data and simulated results, as shown in Table 1. For most cities, $SO_2$ and $PM_{2.5}$ concentrations are underestimated by small margins with NMB in the range of −33.36% to −9.66% and −13.50% to +12.18%, respectively. Compared to those without DMS emissions, the model with DMS emissions increases $SO_2$ concentrations and improves the comparison with observed data by small margins (<3.2%) at all cities. The addition of DMS emissions have a mixed impact on model performance for $PM_{2.5}$. The model without DMS emissions already overpredicts $PM_{2.5}$ at Zhoushan, Ningbo, Qinhuangdao and the addition of DMS emissions marginally deteriorates the comparison with observed data. In other cities, the model with DMS emissions improves the comparison with observed $PM_{2.5}$ concentrations by small margins. The deviations between the simulation and observation occur due to the uncertainties of meteorological fields, emission inventories, chemical mechanisms, and deposition.

**Table 1.** Statistical metrics of $SO_2$ and $PM_{2.5}$ with and without DMS in nine eastern coastal cities of China in 2017.

| City | SO₂ | | | | | | PM₂.₅ | | | | | |
|------|-----|---|---|---|---|---|-------|---|---|---|---|---|
| | Without DMS | | | With DMS | | | Without DMS | | | With DMS | | |
| | NMB (%) | RMSE (μg m⁻³) | r | NMB (%) | RMSE (μg m⁻³) | r | NMB (%) | RMSE (μg m⁻³) | r | NMB (%) | RMSE (μg m⁻³) | r |
| Zhuhai | −10.25 | 10.64 | 0.67 | −9.66 | 10.08 | 0.69 | −8.21 | 21.66 | 0.76 | −5.15 | 18.89 | 0.77 |
| Xiamen | −20.19 | 17.63 | 0.63 | −19.34 | 15.37 | 0.62 | −5.18 | 25.25 | 0.69 | −2.44 | 23.23 | 0.70 |
| Ningbo | −18.69 | 9.36 | 0.70 | −17.26 | 9.05 | 0.70 | 10.23 | 32.80 | 0.72 | 12.18 | 33.32 | 0.70 |
| Zhoushan | −23.76 | 16.66 | 0.75 | −22.33 | 15.52 | 0.73 | 6.15 | 28.94 | 0.65 | 10.22 | 29.62 | 0.65 |
| Shanghai | −20.37 | 10.49 | 0.70 | −18.15 | 10.25 | 0.72 | −7.06 | 32.18 | 0.68 | −2.17 | 28.83 | 0.69 |
| Nantong | −33.36 | 22.72 | 0.64 | −30.05 | 21.44 | 0.63 | −11.23 | 37.06 | 0.73 | −6.67 | 35.42 | 0.75 |
| Qingdao | −20.24 | 14.47 | 0.76 | −19.17 | 14.22 | 0.76 | −6.21 | 29.48 | 0.70 | −2.93 | 28.22 | 0.71 |
| Tianjin | −25.72 | 17.26 | 0.68 | −23.05 | 17.68 | 0.70 | −13.50 | 27.17 | 0.66 | −9.42 | 25.61 | 0.68 |
| Qinhuangdao | −31.47 | 15.32 | 0.77 | −28.26 | 13.13 | 0.79 | 8.67 | 26.41 | 0.70 | 10.55 | 25.58 | 0.68 |

For the innermost Shanghai domain, model predictions with DMS emissions are compared with observed data from three monitoring stations (Pudong Monitoring Station (121.53° E, 31.23° N), Zhoupu Station (121.58° E, 31.11° N), and Lingang Station (121.93° E, 30.91° N)). The monitoring stations are located at distances of 5–22 km from the coast. The observed data of four major atmospheric pollutants ($SO_2$, $NO_2$, $PM_{2.5}$, and $O_3$) are provided by the Shanghai Pudong Monitoring Station. The temporal variations of observed and predicted concentrations in January, April, July, and October in 2017 are shown in Figure S2. The NMB for $SO_2$ is lowest (7.74%) at the Lingang Station, the NMB for $NO_2$ is lowest (10.49%) at the Pudong Monitoring Station, the NMB for $O_3$ is lowest (−20.60%) at the Pudong Monitoring Station, and the NMB for $PM_{2.5}$ is lowest (−8.11%) at the Lingang station. The predicted $O_3$ concentrations are underestimated with NMB ranging between −20.6% to −39.6%. However, the model can reasonably reproduce the observed spatial and temporal trends of these pollutants.

## 3.2. Spatial and Temporal Distribution of DMS Emission Flux

Demonstrated in Figure 3 are the spatial distribution of DMS emission fluxes in four typical months of January, April, July, and October, representing winter, spring, summer, and autumn, respectively. The emission flux of DMS varies considerably in different seasons. The highest flux appears in summer in the East China Sea ranging between 0.02~40.57 µmol m$^{-2}$ d$^{-1}$ and the average flux is 7.42 µmol m$^{-2}$ d$^{-1}$. Autumn has the second highest flux with a maximum of 31.70 µmol m$^{-2}$ d$^{-1}$ and a mean of 4.23 µmol m$^{-2}$ d$^{-1}$. Spring has the third highest flux with a maximum and mean value of 18.22 µmol m$^{-2}$ d$^{-1}$ and 3.70 µmol m$^{-2}$ d$^{-1}$, respectively. The emission flux of DMS is the lowest in winter in the range of 0.52–14.92 µmol m$^{-2}$ d$^{-1}$ with an average of 2.13 µmol m$^{-2}$ d$^{-1}$. Several factors affect DMS emissions; however, the seasonal variations in DMS concentrations in seawater are largely responsible for the variations in the DMS emission flux. For example, the average DMS concentration in seawater in summer is 1.3 times that of autumn, while the average flux of DMS is 1.6 times that of autumn. In addition, the concentration of DMS in seawater in spring is higher than that in autumn, but the sea-to-air emission flux of DMS in spring is lower than that in autumn, which is caused by the higher wind speed in autumn. The total simulated sea area including the Japan Sea is 6.487 million km$^2$. DMS emissions in spring, summer, autumn and winter are 0.13, 0.27, 0.16, and 0.08 Tg S on the basis of the monthly average flux, and thus the estimates of the annual release of DMS is 0.64 Tg S a$^{-1}$.

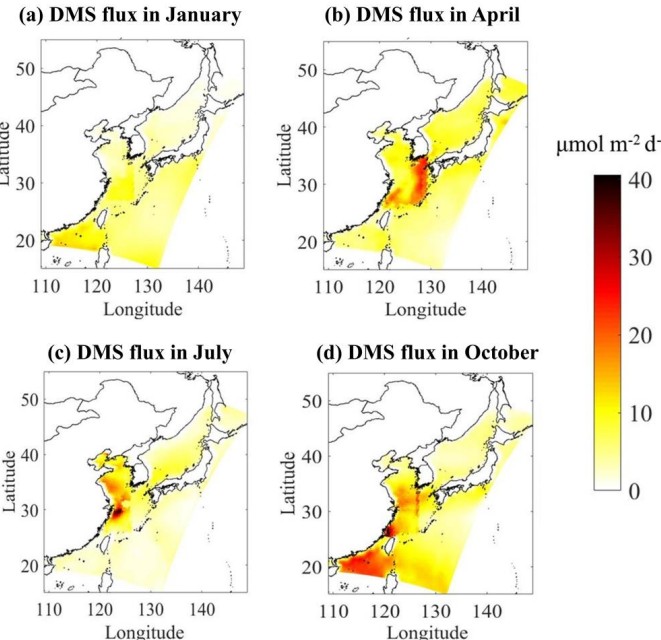

**Figure 3.** The spatial distribution of DMS sea-to-air emission fluxes with LM86 (Liss and Merlivat, 1986) parameterization scheme in four typical months of (**a**) January, (**b**) April, (**c**) July, and (**d**) October.

In order to facilitate the comparison between different sea areas, the DMS flux estimates for the Bohai Sea, the East China Sea, and the Yellow Sea using the localized database are summarized in Table 2. DMS emission flux depends on sea areas and seasons. Even for the same season, fluxes can be quite different among different seas. Taking summer as an example, the emissions fluxes in the Bohai Sea and East China Sea, span two orders of magnitude, range between 0.2–21.1 and 0.1–40.6 $\mu mol\ m^{-2}\ d^{-1}$, respectively. This is the result of large spatial differences in DMS concentration in seawater and wind speed during the simulation period. The above results indicate that geographical location, weather conditions and seasonal changes lead to a large spatio-temporal difference in the air-to-sea emission fluxes of DMS, which will also have a profound impact on global marine DMS flux estimates.

**Table 2.** Seasonal variation of DMS emission flux in different sea areas. (The value outside the parenthesis represents the mean and the ranges inside the parenthesis represent the minimum and maximum, respectively).

| Simulation Period | Bohai Sea | | Yellow Sea | | East China Sea | |
|---|---|---|---|---|---|---|
| | DMS Concentration (nmol/L) | Flux ($\mu mol\ m^{-2}\ d^{-1}$) | DMS Concentration (nmol/L) | Flux ($\mu mol\ m^{-2}\ d^{-1}$) | DMS Concentration (nmol/L) | Flux ($\mu mol\ m^{-2}\ d^{-1}$) |
| January | 0.5 (0.005–1.1) | 2.2 (0.7–4.0) | 1.1 (0.01–2.6) | 2.5 (0.06–6.8) | 1.7 (0.01–2.7) | 3.3 (0.6–10.1) |
| April | 2.7 (0.03–6.0) | 4.2 (0.1–8.7) | 3.7 (0.04–5.5) | 6.6 (0.4–16.7) | 4.9 (0.3–10.7) | 8.8 (0.3–18.2) |
| July | 9.0 (0.02–20.8) | 8.6 (0.2–21.1) | 7.7 (2.4–11.6) | 10.4 (0.4–19.0) | 7.0 (1.3–18.5) | 12.4 (0.1–40.6) |
| October | 2.2 (0.02–5.6) | 5.3 (0.1–10.2) | 3.0 (0.03–4.1) | 8.5 (0.9–14.6) | 3.1 (0.2–5.2) | 10.6 (0.6–26.6) |
| Mean | 3.6 | 5.1 | 3.9 | 6.9 | 4.2 | 8.8 |
| Median | 2.5 | 5.7 | 3.3 | 7.2 | 3.3 | 10.1 |

Based on the simulation results of four typical months, the annual average emission fluxes of DMS calculated by the LM86 method in the Bohai Sea, the Yellow Sea, and the East China Sea are 5.1, 6.9 and 8.8 $\mu mol\ m^{-2}\ d^{-1}$, respectively. Considering the area of the Bohai Sea (about 77,000 $km^2$), the Yellow Sea (about 380,000 $km^2$) and the East China Sea (about 770,000 $km^2$), the annual DMS emissions of the Bohai Sea, the Yellow Sea, and the East China Sea are 0.008, 0.059, and 0.15 Tg S $a^{-1}$, respectively, which is slightly higher than the estimates of 0.074 Tg S $a^{-1}$ in the Yellow Sea and East China Sea by Yang et al. [58] using the LM86 method. The sea area of the Bohai Sea, the Yellow Sea, and the East China Sea account for 0.02%, 0.1%, and 0.2% of the global ocean, respectively, while DMS released into the atmosphere accounts for 0.04%, 0.33%, and 0.8% of the total annual global DMS releases from the ocean.

Accounting for only a small proportion of the global ocean, the Bohai Sea, the Yellow Sea, and the East China Sea are typical continental shelf areas in the world, where the emission flux of DMS is, however, far larger than that of the remote seas in all seasons. The contribution of DMS release from the continental shelf areas is not negligible, which is also consistent with the previous research results [58,59].

### 3.3. Impacts on the Regional Atmospheric Environment

In this study, the localized database and the LM86 method are used to simulate the concentration of pollutants in the atmosphere in the four typical months. The impact of DMS emissions on the atmospheric environment in the eastern coastal region of China is examined by comparing the results with those obtained without DMS emissions.

### 3.3.1. DMS Contribution to $SO_2$ and $SO_4^{2-}$

The $SO_2$, $SO_4^{2-}$ concentrations from the baseline simulation (without DMS) are provided in Figures S3 and S4. Figure 4 shows the absolute contribution of DMS to atmospheric $SO_2$ concentration

in four typical months. The addition of DMS emissions effectively increases $SO_2$ levels in the atmosphere in all seasons. Higher enhancements are predicted over seawater than over coastal areas. In terms of seasons, DMS shows the highest contribution to regional atmospheric $SO_2$ concentration in summer, especially in the Bohai Sea and the East China Sea with a maximum of 0.63 μg/m³, which is consistent with the high simulated DMS emission flux there. In winter, the concentration of DMS in seawater is low, and so is the emission flux, and thus, the average contribution of DMS to the regional atmospheric $SO_2$ concentration is only 0.12 μg/m³.

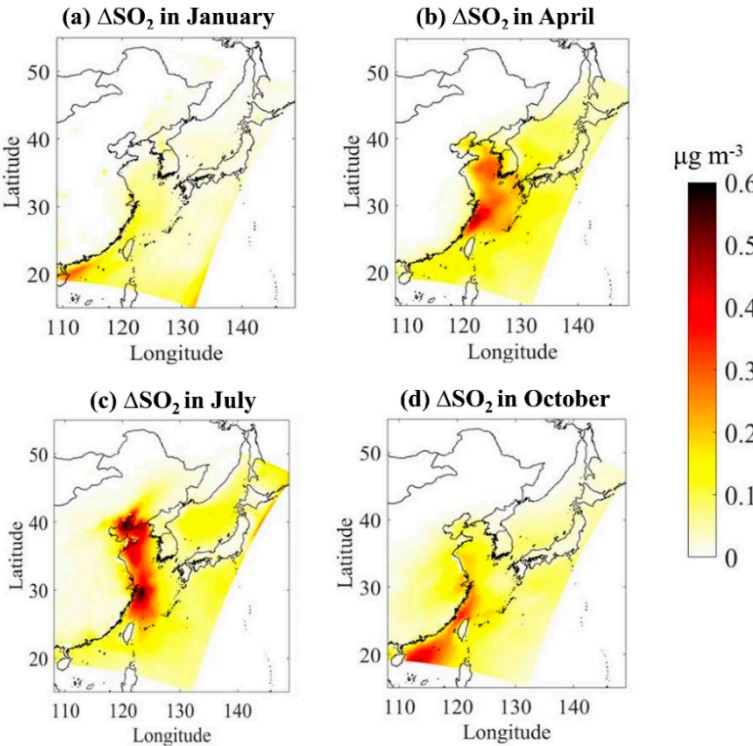

**Figure 4.** Absolute impacts of marine DMS emissions on monthly mean atmospheric $SO_2$ over Chinese seawater in four typical months of (**a**) January, (**b**) April, (**c**) July, and (**d**) October.

Figure 5 shows the absolute contribution of DMS to atmospheric $SO_4^{2-}$ concentration in four typical months. The addition of DMS emissions also increases the atmospheric concentration of $SO_4^{2-}$ over the ocean area by 0.10–0.96 μg/m³. DMS has the highest contribution to the regional atmospheric $SO_4^{2-}$ levels in summer and the highest value occurs in the area from the south of Shandong Peninsula to the northern Jiangsu coast. The contribution to $SO_4^{2-}$ levels is the lowest in winter. The average contribution of DMS in spring, summer, autumn and winter to the atmospheric $SO_4^{2-}$ concentration over the three China Seas (Yellow Sea, Bohai Sea, and East China Sea) is 0.68, 0.52, 0.32, and 0.20 μg/m³, respectively. Unlike $SO_2$, the enhancements in $SO_4^{2-}$ concentration are not limited to the areas with large DMS emission fluxes in the coastal waters of China. It also has a relatively large increase on the land in eastern China and in the open sea. This is mainly due to the shorter residence time of $SO_2$ in the atmosphere, while $SO_4^{2-}$ is mainly distributed in fine-particles and has a longer residence time [60], so it can be transported a longer distance.

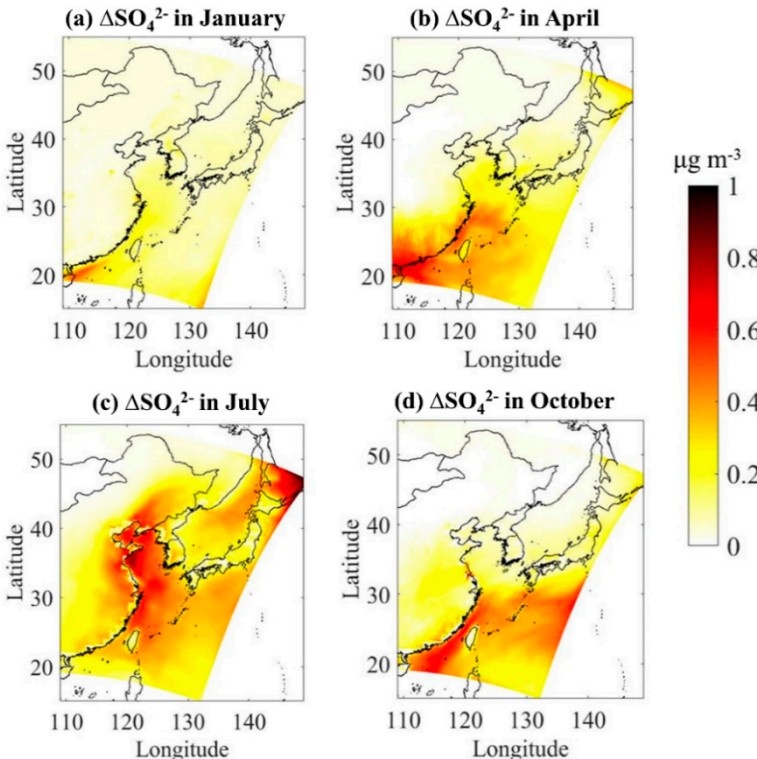

**Figure 5.** Absolute impacts of marine DMS emissions on monthly mean atmospheric $SO_4^{2-}$ over Chinese seawater in four typical months of (**a**) January, (**b**) April, (**c**) July, and (**d**) October.

Overall, the release of marine biogenic DMS from China's offshore regions enhances $SO_2$ and $SO_4^{2-}$ over the ocean, and also affects the coastal land area, but the relative contribution is generally less than 10%, which means the influence of anthropogenic sources still plays a dominant role.

### 3.3.2. Contribution of Marine Biogenic Sulfur Release to MSA and Nss-$SO_4^{2-}$

DMS emitted from the ocean is oxidized into MSA and non-sea salt sulfate (nss-$SO_4^{2-}$) in the atmosphere, and the $SO_2$ produced by the combustion of fossil fuels is also oxidized to eventually generate nss-$SO_4^{2-}$. Therefore, nss-$SO_4^{2-}$ comes from two pathways: DMS released by marine organisms and $SO_2$ emitted by human activities. It is now generally accepted that the only source of MSA in the atmosphere is the oxidation of DMS [29,30]. Therefore, MSA is often used as a tracer to separate $SO_4^{2-}$ of marine biological source from other sources. Figure 6 shows the modeled spatial distribution of atmospheric gas-phase MSA concentration in four typical months. The spatial distribution of MSA concentration in the atmosphere is in agreement with the spatial distribution of DMS fluxes. The average atmospheric gas-phase concentrations of MSA in spring, summer, autumn, and winter are 0.035, 0.056, 0.020, and 0.012 µg m$^{-3}$, respectively, showing significant seasonal differences with higher values in summer and spring and lower ones in autumn and winter. Zhang et al. [61] collected MSA in the atmosphere over the North Yellow Sea in July, 2006 and January, April, and October, 2007. The results showed that the average concentrations of MSA in the atmosphere in spring, summer, autumn and winter are 0.073, 0.039, 0.011, and 0.015 µg m$^{-3}$ with the highest in spring and the lowest in winter, which is slightly different from our results. The observation data of the summer cruise survey in 2018 in the three China Seas show that the concentration of MSA in the atmosphere is in the range of 0.016–0.106 µg m$^{-3}$, with an average of 0.045 µg m$^{-3}$, which is close to our simulation results for the summer of 2017. Perraud et al. [35] turned off the nucleation and uptake into existing particles in the model, left MSA in the gas phase, and reported that the modeling concentration of gas-phase MSA was in the range of 0–0.03 µg/m$^3$, which is basically consistent with our simulation results. It should be noted that the current understanding of atmospheric DMS oxidation pathways is not complete. MSA

can be taken up in aerosols and clouds. Karl et al. [62] noted that liquid-phase oxidation processes are missing in the mechanism. Barnes et al. [63] and recently Chen et al. [64] reported that the multiphase chemistry of DMS is needed for accurately calculating atmospheric MSA production. Our chemical scheme does not incorporate these pathways; thus, future studies are needed to improve the DMS chemistry and the MSA predictions.

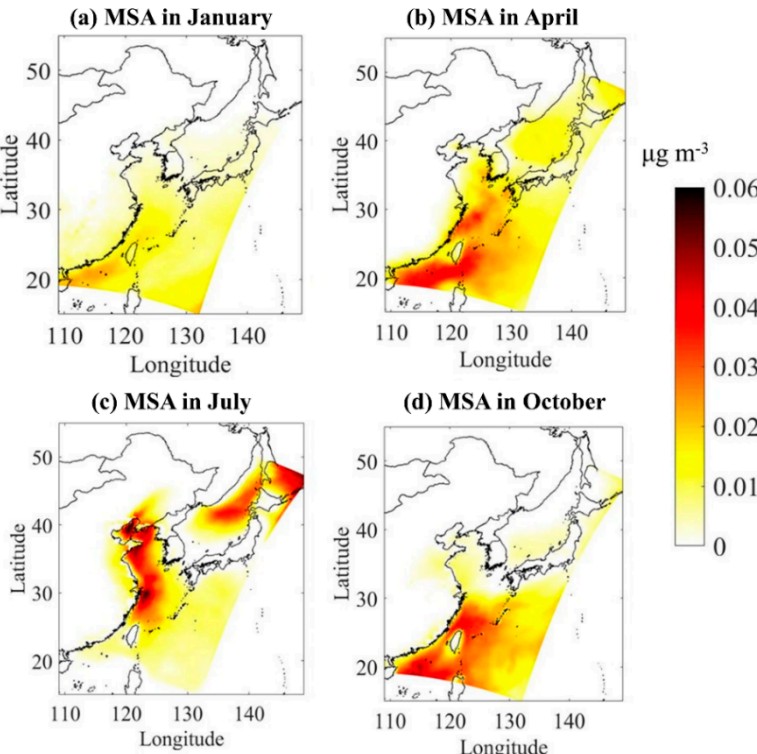

**Figure 6.** The spatial distribution of atmospheric gas-phase MSA in four typical months of (**a**) January, (**b**) April, (**c**) July, and (**d**) October.

It can be concluded from Section 3.3.1 that the absolute difference in $SO_4^{2-}$ concentration between the case with and without DMS emissions is the concentration of biogenic sulfate (nss-$SO_4^{2-}$ bio) contributed by marine biogenic DMS. The calculated ratios of nss-$SO_4^{2-}$ bio/MSA range between 12–17, which are lower than the values of 18–20 reported by Savoie et al. [65] for the clean marine atmosphere. The ratio of nss-$SO_4^{2-}$ bio/nss-$SO_4^{2-}$ in the eastern sea area of China, that is, the average contribution of DMS to total nss-$SO_4^{2-}$ is 7.7% in spring, 4.3% in summer, 3.0% in autumn, and 1.2% in winter, respectively. The observation results by Zhang et al. [61] in the North Yellow Sea show that DMS in spring, summer, autumn, and winter accounts for 11.0%, 10.4%, 2.0%, and 2.8% of the total nss-$SO_4^{2-}$, respectively, with an average value of 6.6% ± 4.8%. Nakamura et al. [66] estimated that the contribution of DMS to nss-$SO_4^{2-}$ is 0–38% in the East China Sea, with an average value of 7.9%. Our results are consistent but slightly lower than long-term observations [67] and model simulations [68] in the Eastern Mediterranean, where marine biogenic DMS accounts for about 17% of nss-$SO_4^{2-}$ in summer and less than 10% in winter. The Eastern Mediterranean region is surrounded by land and greatly affected by human activities which is similar to the situation in China's offshore waters. The results reveal that although contributions of anthropogenic emissions to the marine atmosphere are predominant, the DMS emissions in China's waters are also a non-negligible source especially in spring and summer.

## 3.4. Impacts on the Urban Atmospheric Environment

Due to the confluence of marine air masses and polluted air from coastal cities, the atmospheric environment of coastal cities will also be affected by marine biogenic DMS emissions. Therefore, under

the condition of reduced anthropogenic sulfur emissions, the contribution and role of DMS emissions from polluted sea areas to the atmospheric environment of coastal cities should be reassessed.

3.4.1. DMS Contribution to $SO_2$ and $SO_4^{2-}$

The absolute and relative contribution of DMS to the annual average concentration of $SO_2$ and $SO_4^{2-}$ in Shanghai is shown in Figure 7. For $SO_2$, higher values appear in the Yangtze River estuary with contribution of 0.14 to 0.16 µg/m$^3$ (about 4 to 8%). The $SO_2$ concentration is weakly affected by DMS in the Shanghai city area near the Yangtze River estuary. Except for Baoshan and some urban district areas, where the contribution is ~0.13 µg/m$^3$ (about 4%), the contribution of DMS to the $SO_2$ levels in Shanghai is mostly around 0.08 µg/m$^3$ (about 2%). The higher impact on $SO_4^{2-}$ concentration is found on the southeast boundary of Shanghai, with a contribution of 0.12 µg/m$^3$ (about 10%). Compared with $SO_2$, the impact of DMS on the $SO_4^{2-}$ levels in Shanghai urban areas is more uniformly distributed, with a contribution of 0.06 µg/m$^3$ (about 5%). It should be noted that NOx emissions and hence $NO_3$ radical concentrations are relatively low in clean, remote areas of the ocean. Thus, DMS is primarily oxidized by the OH initiated pathway in such areas. In contrast, NOx emissions and hence $NO_3$ radical concentrations are higher in coastal areas with anthropogenic pollution. Thus, the $NO_3$ pathway plays an important role in DMS oxidation in such environments. Both the day- and night-time oxidations of DMS are important in coastal areas with anthropogenic pollution while the daytime oxidation is the most important pathway in clean remote areas.

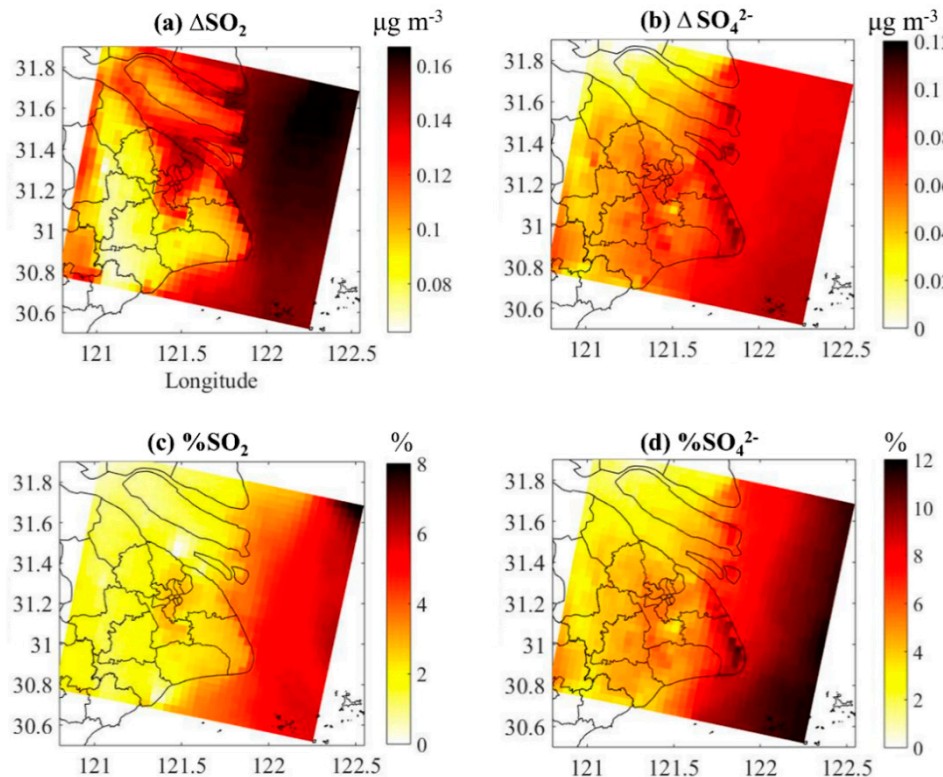

**Figure 7.** The absolute contribution of DMS to the annual average concentration of (**a**) $SO_2$ and (**b**) $SO_4^{2-}$, and the relative contribution of DMS to the annual average concentration of (**c**) $SO_2$ and (**d**) $SO_4^{2-}$ in Shanghai.

3.4.2. DMS Impact on $O_3$

The absolute and relative contribution of DMS to the annual average concentration of $O_3$ in Shanghai is shown in Figure 8. The addition of DMS to the model reduces surface $O_3$ concentration. This effect is relatively higher in the sea area with a reduction of about 0.65~0.90 ppbv (about

1.5–2.5%). The impact of DMS emissions on the urban land area of Shanghai is relatively small, where the $O_3$ concentration is reduced by 0.5 ppbv (about 1.25%) at the coastal line and 0.25~0.40 ppbv (about 0.5 to 1%) in most urban areas. The reactions of DMS with iodine monoxide (IO) and bromine monoxide (BrO) release iodine (I) and bromine (Br) atoms which react with $O_3$ to reduce its concentration. In addition to the contribution to aerosols, marine biogenic DMS emissions also influence $O_3$ which can be an area of further research.

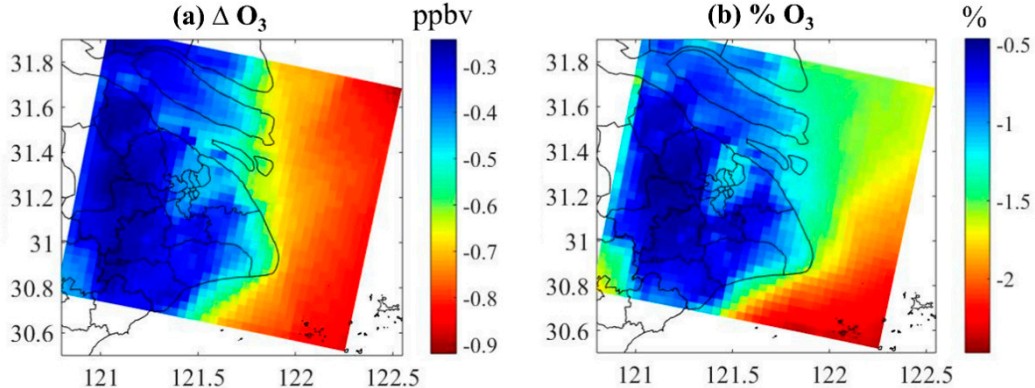

**Figure 8.** The (**a**) absolute and (**b**) relative contribution of DMS to the annual average concentration of $O_3$ in Shanghai.

### 3.4.3. Effect on the Relative Content of Fine-Mode Aerosol Mass Components

We calculate the relative amount of fine-mode aerosol in total aerosol (fine- and coarse-mode) and present the results in Table 3. Without DMS, the percent of $NO_3^-$ in fine-mode aerosol is quite different in the two airsheds. In the urban atmosphere of Shanghai, 16% $NO_3^-$ aerosol is found in the fine-mode while only 5.2% of $NO_3^-$ is present in the fine-mode in the marine atmosphere. This is likely due to the high removal rate of nitric acid in the marine air which contains relatively high concentrations of coarse alkaline sea salt particles. The fine-mode $NH_4^+$ and nss-$SO_4^{2-}$ is similar in the urban and the marine atmospheres with an average content of 75% and 74% in the urban air, respectively (without DMS). Consistent with the results of Berresheim et al. [69], our analysis shows that aerosol nss-$SO_4^{2-}$ and $NH_4^+$ remain mostly in the fine-mode, while $NO_3^-$ mainly resides in the coarse-mode. The addition of DMS emissions marginally affects the fine-mode $NO_3^-$ content in the urban and marine atmosphere (<1%). However, the addition of DMS increases fine-mode nss-$SO_4^{2-}$ by 4% and 9% in the urban and marine atmosphere, respectively. It also increases $NH_4^+$ in the fine-mode by 4–5% in the urban and marine atmospheres. The study of Gross et al. [70] also showed that under clean marine atmospheric conditions, DMS can increase nss-$SO_4^{2-}$ of the total accumulation mode by 5–15%.

**Table 3.** Relative amount of fine-mode aerosol in total aerosol (fine- and coarse-mode) mass in urban and marine atmospheres.

| Aerosol Compounds | Urban Air | | Marine Air | |
| --- | --- | --- | --- | --- |
| | Without DMS Emissions | With DMS Emissions | Without DMS Emissions | With DMS Emissions |
| $NO_3^-$ | 16% | 15% | 5.2% | 5.0% |
| $NH_4^+$ | 75% | 79% | 70% | 75% |
| nss-$SO_4^{2-}$ | 74% | 78% | 67% | 76% |

## 4. Conclusions

A national scale database of DMS concentration in China seawater is constructed by combining the DMS measurements of historical cruise surveys from 2009 to 2017 in China's continental shelf waters with DMS concentrations in seawater from the global database. The sea-to-air emission flux

of DMS is calculated based on the database and the LM86 parameterization, which varies greatly in different seasons with the highest fluxes in summer and the lowest in winter. The DMS emissions in the entire simulated sea area for spring, summer, autumn, and winter are 0.13, 0.27, 0.16, and 0.08 Tg S, respectively. The annual mean fluxes of DMS in the Bohai Sea, Yellow Sea, and East China Sea are 5.1, 6.9, and 8.8 $\mu$mol m$^{-2}$ d$^{-1}$, respectively, and the annual DMS releases can be estimated 0.008, 0.059, and 0.15 Tg S a$^{-1}$, respectively. Although the continental shelf area accounts for a small proportion of the global ocean, the contribution to the global ocean DMS release cannot be ignored.

At the regional scale, the addition of DMS emissions increases the concentration of $SO_2$ in the atmosphere of the eastern coastal region of China, with the highest contribution in summer (0.63 $\mu$g/m$^3$) and lowest in winter (0.12 $\mu$g/m$^3$). The addition of DMS emissions also increases $SO_4^{2-}$ levels by 0.52 $\mu$g/m$^3$ in spring, 0.68 $\mu$g/m$^3$ in summer, 0.32 $\mu$g/m$^3$ in autumn and 0.20 $\mu$g/m$^3$ in winter, respectively. Higher enhancements are predicted not only over seawater but also over coastal areas. The ratio of nss-$SO_{4\ bio}^{2-}$/nss-$SO_4^{2-}$ in the eastern coastal of China, that is, the average contribution of DMS to total nss-$SO_4^{2-}$ is 7.7% in spring, 4.3% in summer, 3.0% in autumn, and 1.2% in winter, respectively, indicating the non-negligible role of DMS emissions to the marine atmosphere.

At the urban scale, the contribution of DMS to the $SO_2$ levels in Shanghai is mostly around 0.08 $\mu$g/m$^3$ (about 2%). In contrast with $SO_2$, the impact of DMS on the $SO_4^{2-}$ concentration in Shanghai urban areas is more uniform, with an average contribution of 0.06 $\mu$g/m$^3$ (about 5%). The model with DMS reduces the concentration of $O_3$ in the atmosphere. The reduction of $O_3$ is relatively larger in the open sea than in the urban area of Shanghai. The addition of DMS also increases fine-mode $NH_4^+$ distribution by 4% and 5%, and nss-$SO_4^{2-}$ by 4% and 9% in the urban and marine atmosphere, respectively. Future studies are needed to investigate the climate impact of DMS emissions in China.

**Supplementary Materials:** The following are available online at http://www.mdpi.com/2073-4433/11/8/849/s1. Figure S1: DMS concentration in seawater from the Global Surface Seawater DMS in different seasons. (http://saga.pmel.noaa.gov/dms/), Figure S2: Comparison of observations and simulation results of four major atmospheric pollutants ($SO_2$, $NO_2$, $O_3$ and $PM_{2.5}$) at Pudong Station, Zhoupu Station and Lingang Station in Shanghai, Table S1: Summary of historical cruise surveys for DMS concentration in seawater.

**Author Contributions:** Conceptualization, Y.Z.; Data curation, S.L., J.Z. and S.Z.; Funding acquisition, Y.C. and G.Y.; Methodology, Y.Z. and G.S.; Project administration, Y.C. and G.Y.; Software, J.Z. and G.S.; Visualization, S.L.; Writing—original draft, S.L.; Writing—review & editing, Y.Z., G.S. and A.S.-L. All authors have read and agreed to the published version of the manuscript.

**Funding:** This work was supported by the National Key Research and Development Program of China (grant no. 2016YFA060130X) and the National Natural Science Foundation of China (Grant No. 21677038) and the Major Program of Shanghai Committee of Science and Technology, China (No.19DZ1205009).

**Acknowledgments:** We want to thank the Ocean University of China for the DMS observation data and U.S EPA for the internal review and valuable suggestions. The views expressed in this paper are those of the authors and do not necessarily represent the views or policies of the U.S. EPA.

**Conflicts of Interest:** The authors declare no conflict of interest.

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
