# Peer review of "Regional and Urban-Scale Environmental Influences of Oceanic DMS Emissions over Coastal China Seas"

_atmosphere, doi:10.3390/atmos11080849_

Round 1
Reviewer 1 Report
The article “Regional and urban-scale environmental influences of oceanic DMS emissions over coastal China seas” by Shanshan Li et al., code: atmosphere-864105 assesses the impact of oceanic DMS emissions on the atmospheric composition on the regional scale over coastal eastern China and on the urban scale in the area of Shanghai. The WRF-CMAQ model is employed in version 5.2, with added halogen chemistry and DMS chemistry. Apart from the expected results of increased SO2 and sulfate, interesting results about ozone reduction in Shanghai and influences on the distributions of fine-mode ammonium and coarse-mode nitrate in the marine and urban atmosphere are presented. The manuscript is overall well organized and all used data sources are properly referenced in the text, except for the emissions of halogens from seawater. The authors should be aware that considerable uncertainties still persist in the understanding of the oxidation routes of DMS in the atmosphere, with consequences for both air quality assessment and climate predictions. From this perspective, additional sensitivity calculations with the CMAQ model should be performed that shed light on the relevance of the different oxidation pathways (OH, NO3, halogen radicals, halogen oxides) for the predicted SO2 and sulfate concentrations in the marine and urban atmosphere. In addition to these more general considerations to improve the paper, several specific issues need to be addresses before I can recommend publication in the journal Atmosphere:
- L89: “Sarwar et al. [34] incorporated halogen chemistry …” – does it mean that the oxidation of DMS by halogen radicals and halogen oxides was considered in that study? If yes, it should be mentioned in the Introduction how much ozone was affected by addition of DMS chemistry.
- The work of Perraud et al. (2015) [www.pnas.org/cgi/doi/10.1073/pnas.1510743112] using the University of California Irvine–California Institute of Technology airshed model should be cited in the context of regional-scale simulation of DMS chemistry in coastal urbanized areas. That study also assessed DMS emissions from agricultural activities.
- DMS chemistry is not included in the documentation of the CMAQ v5.2 model and as far as I know, it is only available in CMAQ v5.3. I could not find DMS oxidation reactions in any of the chemistry mechanisms that are part of the distribution of CMAQ v5.2. The name of the used chemistry mechanism should be stated explicitly. A table of the considered DMS oxidation reactions must be included in the manuscript or the supplementary materials.
- Which method was used to spatially interpolate or extrapolate the seawater DMS data obtained from the cruise surveys (L169-L172)? Explain in more details.
- An explanation of the seawater emissions of halogens is missing. Geographic maps of the halogen emissions should be included either in the manuscript or in the supplementary materials.
- There is a mistake in Equation (5) of the water-side transfer velocity. Note that kw is proportional to Sc^(-2/3) for low wind speed (U10 < 3.6 m/s) and proportional to Sc^(1/2) for higher wind speeds. Thus for U10 < 3.6 m/s, the denominator has to be (ScDMS/600)^(2/3). Please check if this mistake affected the calculated DMS fluxes.
- Section 3.2.2: the predicted nss-SO42-bio/MSA ratios are substantially affected by the uncertainties in DMS chemistry. In particular, the mechanism of production of MSA from DMS gas phase oxidation by hydroxyl radicals (at daytime) and nitrate radicals (at nighttime) remains unclear, as outlined in Karl et al. (2007) [Karl M, Gross A, Leck C, Pirjola L (2007) Intercomparison of dimethylsulfide oxidation mechanisms for the marine boundary layer: Gaseous and particulate sulfur constituents. J Geophys Res 112(D15):D15304.]. Although simulation results for MSA concentrations and its seasonal variation appear to be in agreement with observations from the China seas, they are not necessarily matching for the right reason, because MSA condenses to aerosol particles and can be taken up in the liquid phase of aerosols and clouds. A discussion of the uncertainties in the DMS oxidation scheme referring to the study by Karl et al. (2007) should be added here. Please give an estimate of the model uncertainties of the gas-phase MSA production.
- In coastal regions with anthropogenic pollution, the DMS oxidation routes are changing compared to the clean marine atmosphere. With higher levels of NOx, the NO3 radical concentrations at nighttime can reach several 10^8 cm-3, rendering the oxidation of DMS by NO3 radicals the overall dominant pathway. The relevance of this change in oxidation pattern when approaching the urban scale should be discussed based on sensitivity calculations (as already mentioned in my general comments).
- Section 3.4: did the inclusion of DMS emissions improve the predictions of SO2 at the monitoring sites? Please add the bias, error and correlation coefficient from the comparison with the run without DMS emissions in Table 3.
Technical corrections:
L85 and L102: Please replace “numerical model” by “numerical modeling”.
Figure 1: In the zoomed map of D03, the coverage by the urban areas (urban land cover) should be marked.
Figure 2: The areas that are not covered by the observations should be indicated in the maps.
Table 1: It need to be stated in the table caption what the value ranges in brackets are representing. A left round bracket is missing for the units of flux from Bohai Sea. The median values should be added in addition to the mean values.
L238-L239: “Lana et al. [13] reported …” – this was already mentioned in the Introduction; the sentence can be deleted.
L245: Better use “remote” instead of “distant”.
Figure 6 and associated text: I assume that atmospheric MSA always denote the MSA in the gas phase. This should be stated clearly, for example in the caption of Figure 6, as “atmospheric gas-phase MSA”.
L424-L425: What are the given two percentage values referring?
Reviewer 2 Report
The study by Li et al. uses a regional air quality model to simulate the contribution of ocean DMS to air quality relevant atmospheric consituents at the local and regional scale over eastern China. The study finds modest contributions of DMS to SO2 and SO4, the marine source being dwarfed by the anthropogenic sources. The contribution to aerosol loading also appears to be small, but not insignificant. Marine DMS induces a small decrease in O3 concentration.
The manuscript is mostly well-organised and written. The results are as expected, and are a useful indication of the potential role of DMS in polluted environments.
I have a few comments which I believe should be addressed before publication.
- The abstract refers to both absolute (ug.m-3) and relative (%) contributions for different species. For clarity, referring to only relative contributions would be useful
- The chemistry scheme is briefly described, and is well known. However, no information is provided about the aerosol scheme, AERO6. Where is the model description reference? What are the basic characteristics of the scheme?
- Lines 155 and 156 refer to ‘tons’. Presumably these are metric tons? However, for the sake of clarity, it would be preferable to use kg, Tg, etc
- Eq. 1 and line 191. Is it ‘Cg * H’, or ‘Cg/H’?
- Figures 4 and 5 provide absolute differences, while the text provides relative differences (e.g. line 280). Please provide maps (perhaps in the supplementary material) of SO2, SO4 etc concentrations from the baseline simulation (without DMS). Doing so provides a more complete overview to the reader.
- Please provide another figure containing the contribution of DMS to PM2.5, similar to Figures 4 and 5. PM2.5 is an important quantity and its omission is obvious.
- Section 3.4 is important to establish that the model works, and should be included prior to the main results. Once the validity of the model is established, it is then possible for the reader to proceed to the main part of the paper, knowing the strengths and weaknesses of the model
Round 2
Reviewer 1 Report
The revised manuscript considers my previous concerns in adequate manner.
I can now recommend publication in Atmosphere.
